# Farm Animal Welfare—From the Farmers’ Perspective

**DOI:** 10.3390/ani14050671

**Published:** 2024-02-21

**Authors:** Clive J. C. Phillips

**Affiliations:** 1Curtin University Sustainability Policy (CUSP) Institute, Kent St., Bentley 6102, Australia; clive.phillips@curtin.edu.au; 2Institute of Veterinary Medicine and Animal Sciences, Estonian University of Life Sciences, Kreutzwaldi 1, 51014 Tartu, Estonia

**Keywords:** animal welfare, farmer, attitude, resource cost, feed, climate change

## Abstract

**Simple Summary:**

Considerations of farm animal welfare are usually driven by either what is important to the animal or the consumer. Farmers also play a key role in farm animal welfare as they manage the interfaces between, first, the animal and its environment and, second, the supply of animals to the retailer and thence to the consumer. This commentary seeks to put farmers’ perspectives on what is important in animal welfare and what options they have to address the key issues. The most important issues are identified as climate change, availability and cost of key resources, social licence, antibiotics and disease, long-distance transport and biodiversity loss. Options for them to diminish adverse effects on welfare include improved nutrition, diversification to other enterprises, silvopastoral systems, reversing genetic gains and precision livestock farming. It is concluded that farmers play a vital role that should be recognised as a key component in determining the welfare of farm animals.

**Abstract:**

Improvements in the welfare of animals in the intensive production industries are increasingly being demanded by the public. Scientific methods of welfare improvement have been developed and are beginning to be used on farms, including those provided by precision livestock farming. The number of welfare challenges that animals are facing in the livestock production industries is growing rapidly, and farmers are a key component in attempts to improve welfare because their livelihood is at stake. The challenges include climate change, which not only exposes animals to heat stress but also potentially reduces forage and water availability for livestock production systems. Heat-stressed animals have reduced welfare, and it is important to farmers that they convert feed to products for human consumption less efficiently, their immune system is compromised, and both the quality of the products and the animals’ reproduction are adversely affected. Livestock farmers are also facing escalating feed and fertiliser costs, both of which may jeopardise feed availability for the animals. The availability of skilled labour to work in livestock industries is increasingly limited, with rural migration to cities and the succession of older farmers uncertain. In future, high-energy and protein feeds are unlikely to be available in large quantities when required for the expanding human population. It is expected that livestock farming will increasingly be confined to marginal land offering low-quality pasture, which will favour ruminant livestock, at the expense of pigs and poultry unable to readily digest coarse fibre in plants. Farmers also face disease challenges to their animals’ welfare, as the development of antibiotic resistance in microbes has heralded an era when we can no longer rely on antibiotics to control disease or improve the feed conversion efficiency of livestock. Farmers can use medicinal plants, pro-, pre- and synbiotics and good husbandry to help maintain a high standard of health in their animals. Loss of biodiversity in livestock breeds reduces the availability of less productive genotypes that survive better on nutrient-poor diets than animals selected for high productivity. Farmers have a range of options to help address these challenges, including changing to less intensive diets, diversification from livestock farming to other enterprises, such as cereal and pseudocereal crops, silvopastoral systems and using less highly selected breeds. These options may not always produce good animal welfare, but they will help to give farm animals a better life.

## 1. Introduction

Approaches to animal welfare usually focus on either the animal’s perspective [1] or that of the public [2], as both are crucial components in determining welfare. However, the perspectives of farmers, who are largely responsible for the interfaces between, first, the animal and its environment and, second, the interface between the animal and retailers (and, hence, consumers), are largely ignored. Farmers tend to have a different concept of animal welfare to the public, in which they consider the long-term effects of good feeding, shelter, health management and husbandry as more important than procedures that cause animals pain in the short term [3,4]. They are especially more inclined to focus on animals’ health and less on their environment [5]. By contrast, animal advocate members of the public tend to rate injurious practices as more important, such as castration, dehorning and preslaughter stunning [4]. This could be because farmers, as managers of the interface between the environment and their animals, understand animal welfare well, but it is more likely to be because farmers become inured to the impact of injurious practices on their animals [6]. 

This commentary considers the important challenges that farmers currently face that will impact on the welfare of their animals. Farmers’ ability to manage the welfare of their animals depends on many factors, including their economic circumstances, extraneous influences on their farms, such as the weather, their time availability to implement practices relevant to welfare and the resources at their disposal [7]. Many of these are well known, with a clear impact on welfare, but new challenges are also appearing, which are considered in the next section. 

## 2. New Challenges for Livestock Farmers

### 2.1. Climate Change

Livestock farmers are being severely impacted by anthropogenic climate change [8,9]. Not only are global temperatures increasing, the magnitude of the El Nino Southern Oscillation has been increasing since the 1950s, as well as other extreme weather events [10,11,12]. Livestock experience heat stress but, unlike some other disorders, e.g., lameness [13], there is little or no information on the ability of farmers to recognise heat stress in their livestock or whether they will act if they do detect it. Heat stress adversely affects animals’ biological functioning, but the extent of this negative effect is unclear [14,15]. 

High temperatures and droughts induced by anthropogenic climate change also cause reductions in the quantity and quality of feed and water, premature deliveries of offspring and more rapid spread of livestock diseases and direct heat stress effects on the animals, including increased water consumption and decreased feed conversion efficiency [10]. Droughts are at least partly behind the increased livestock price fluctuations in recent years, with high prices during drought years and low prices during and after wet years, which make it hard for livestock farmers to plan their enterprises. Effects on pasture production are varied: negative in arid and semiarid zones and positive in humid temperate regions. Forage may have increased cellulose and hemicellulose contents, reducing digestibility [16]. The reduction in water availability will particularly affect dairy cattle and feedlot cattle in arid zones, whose water requirements are directly related to ambient temperature [17]. Furthermore, heat stress induced by climate change reduces the productivity of beef and dairy cattle and increases transport mortality. Most livestock have an upper critical temperature (UCT) of 25–30 °C (except neonates, which have UCTs in the region of 35 °C) [18]. Above the UCT, physiological adaptations are invoked to assist the animal in coping with the heat stress, leading to reduced efficiency of production and, if not successful, continued stress and ultimately death. In most livestock-producing regions of the world, the risk of heat stress is significant during hot periods of the year. Animals most at risk are grazing livestock because of direct exposure to solar radiation [19] and those in buildings with low roofs that transmit heat [20]. 

Reproduction is also adversely affected, with short oestrus periods in females, often displayed at night, and premature parturition that reduces the viability of offspring [21]. Milk production of dairy cows declines by up to 50% in heat stress, with reductions in milk quality, and the growth of beef cattle and meat chickens also declines [22]. Similarly, the quality and quantity of eggs from layer hens declines [18]. These effects have been well known for several decades but accurate predictions of effects of climate change on individual countries are now urgently needed. For example, taking the effects of the growing human population and increasing temperatures together, it is predicted that milk availability in a hot country, Egypt, will decline from 61 kg/person per year in 2011 to 26 kg/person per year in 2064 [22]. 

### 2.2. Availability of Feed and Other Resources

As the world’s human population grows up until the end of this century, less cereal grain will be available for feeding to livestock, which is currently about 36% of total grain production [23]. Increasingly, livestock will have to utilise marginal land—land that is too dry, steep, inaccessible or far from human populations—or rely on feedstuffs that are byproducts of human food production. This is expected to favour ruminant livestock at the expense of pigs and poultry, which are unable to digest as effectively the coarse fibre in herbage, e.g., [24] that grows in such places. However, although this might seem logical in an under-supplied cereal market for both animals and humans [25], inequalities of purchasing power will probably prolong intensive livestock production. Wealthy consumers will attempt to continue to support these industries, which are increasingly likely to be in developing countries because of these countries’ need for foreign currency and less concern for animal welfare [26,27]. This scenario, in which inefficient and polluting industries are maintained in developing countries to feed consumers in the rich northern hemisphere countries, should be prevented by global agreement on the need for reform of livestock production industries [28]. 

Associated with rapidly changing climate and the difficulties this presents for farmers are the rapidly escalating costs of feed for livestock farmers. Feed costs escalated dramatically during the COVID-19 pandemic [29] due to distribution issues but this period also coincided with drought in Australia. Climate change principally affects feed supply through limiting water availability but, also, arid regions could become more saline [30]. Plant pests and diseases are likely to multiply, and extreme weather events may support novel pathogens. Smallholders are most vulnerable as they do not have resources to save crops or seed stocks [31] and often have few spare resources, with the result that their feed stocks and animal welfare are likely to be most at risk. 

### 2.3. Declining Human Capital

Human capital is an essential part of profitable livestock production systems. There continues to be both a migration to cities and away from rural areas and from the less developed and livestock dependent countries in the south towards the industrialised countries of the northern hemisphere [32]. This makes providing skilled labour to care for animals more difficult, with the inevitable result that animals’ welfare deteriorates. As well, recruitment into the agricultural industries is difficult at times of increasing labour shortages, and the question of who will succeed older farmers is increasingly uncertain. As a result, local knowledge of how best to care for farm livestock may be lost [33]. Every farm is different, even from neighbouring farms, and, in the past, the knowledge of how to best utilise the resources that the farm provides was passed from one generation to the next. This is no longer assured, and animal welfare may suffer as a consequence. 

With the industrialisation of livestock farming, there is a need for the labour force to be increasingly skilled. An important asset of skilled labour is the ability to detect behavioural abnormalities in the livestock they care for. For example, dairy farmers may have difficulty detecting cows that are lame within their herd, which is usually by observation by farmers and their staff [13,34]. The detection of lame cows is increasingly difficult as herd sizes increase, and staff have less time to detect abnormalities. Automated detection methods are being developed but their cost, effectiveness, and negative farmer perception have led to limited uptake [34]. Availability of veterinary attention also cannot be guaranteed, as the costs of treating individual animals may exceed their value [35]. 

### 2.4. Economic Constraints

The price of both raw material inputs and products of animal production systems has experienced growing volatility since the year 2000 because of currency fluctuations, spread of diseases such as African swine fever, use of feed grain for biofuel production, labour shortages, the war in Ukraine, and many other factors [36,37]. This makes it risky for livestock farmers to invest in facilities that will provide good animal welfare. For example, there has been pressure on poultry farmers to stop keeping chickens in small cages but investment in either larger cages with more resources for the birds or in free-range systems is difficult when faced with such uncertainty. Smallholders face the biggest problems, as they often have low incomes and such investment is often not feasible [38], especially given the reluctance of banks to finance farm development in uncertain financial markets. Investment in animal welfare requires stable markets, assured returns to producers, and confidence that consumers will pay a higher price for products from animals with better welfare.

Reduced availability of cereal grains will create poor welfare in some livestock production systems. Dairy cows rely on nutrient-dense cereals to consume sufficient energy and protein to produce the high volumes of milk that they have been bred to produce [39]. Without them, they will attempt to support high milk yields by mobilising body tissue, especially in early lactation. Chronically underweight cows will have little energy for normal activity, may feel chronic hunger, and will be unlikely to conceive. Pigs and chickens also currently rely on cereal grains for high growth and egg production, and they consume approximately 70% of animal feed grains worldwide [40]. Without these feeds, animal growth would be slow and pork and egg production much diminished. As meat chicken production systems have grown faster than any other animal production sector, there should be serious concern about the ability to maintain this sector in the future.

Fertilisers have been applied to animal feed crops at high rates for several decades in an attempt to maximise the carrying capacity of farmers’ land. Any threat to this high output of feed crops risks undernutrition of the farmers’ livestock and, consequently, poor welfare. Fertiliser prices have shown the greatest increases since the year 2000, mainly due to increased cost of raw materials and natural gas, increased transport costs, and a shortage of supply with the growing world population’s need for increased food production [41]. Phosphates are likely to run out within a few decades [42]. The high fertiliser prices have caused many smallholder farmers to be both feed- and food-insecure [43].

### 2.5. Antibiotics and Disease

Good animal welfare is currently maintained in intensive animal production by antimicrobials, and stopping antibiotic use will have negative effects on welfare. Welfare will be very significantly affected when and if antimicrobials are finally removed from the pig and cattle industries. The threat of disease is ever-present, and the impact of new diseases, such as African swine fever (ASF), on welfare is catastrophic. In a period of just 13 years, ASF has seriously affected most of the world’s porcine industries [44]. The disease itself has major welfare implications and mortality rate is high [45]. Because an intermediate host, wild boar, is being controlled as a result of the disease, there are also major welfare issues associated with the cull, which might be considered to be a collateral animal welfare impact. Similar considerations exist for avian influenza, now widespread in the northern hemisphere and threatening South America. Wild bird populations are also being badly affected. 

Antibiotics have also been used in the intensive livestock industries to improve digestion efficiency for at least 80 years. Over this period, it was inevitable that bacteria would develop resistance to the antibiotics because of their rapid replication rate. As antibiotics have also had a major role in controlling transmissible human disease, their use in the intensive livestock industries has been banned in the European Union and several other developed countries [46]. However, with intensive livestock production growing rapidly in developing countries, it is of no surprise that, in the current decade, it is expected that antibiotic use will *increase* by 8%, mainly in Asia and South America, due to the rapidly growing intensive livestock industries there [47]. Approximately 73% of all antimicrobial drugs are used in animals [47]. The use of antibiotics in poultry, which is largely to improve digestive efficiency, has declined but that in pigs and cattle, which is partly to control disease, has remained constant or increased [47]. Smallholders in Africa still have significant challenges from poultry diseases, such as *E. coli* and salmonella [48]. 

### 2.6. Transport

Not all farmers are aware that transport represents one of the biggest welfare challenges to livestock, especially when it is over long distances [49], as they are not usually directly involved. Travel such as the 1249 km sea voyage for cattle from Patagonia to central Chile is an example of such a welfare challenge [50]. Inadequate feed and water supplies cause undernutrition and dehydration and the high stocking density exacerbates the problem, as not all cattle are able to gain access to feed and water. Stocking densities should relate to the animals’ needs for standing up, lying down, turning around, and movement to feed and water supplies [51]. Failure to meet these basic needs results in aggression between animals and an inability of animals that are lying down to get up [52]. The need to take the cattle by this route in Chile is created by the lack of local abattoir facilities and, worldwide, the reduction in the number of abattoir facilities because of stringent meat hygiene and slaughter standards has meant that livestock have to travel further. In transport by trucks, the absence of feed and water causes significant stress, but providing these can create a dusty environment in the case of feed or a wet, slippery floor in the case of water. Additional to the stress caused by shortages of feed and water, a novel environment, very close proximity of conspecifics, motion sickness, and loss of balance in response to vehicular motion [53] all contribute to a stressful experience. As well as transportation to slaughter, farmers are increasingly aware that transport between farms can transfer diseases. This is increasingly common with the specialisation of livestock farms for single purposes, e.g., producing youngstock. 

### 2.7. Loss of Biodiversity

Livestock breeds have been developed over the last 70 years that grow faster, produce more eggs, increase muscle growth, and have larger litters. Traditional livestock breeds are fast disappearing [54], which may cause problems if they are needed when large quantities of cereal grain are not available to feed to the high-output livestock. Without this feed, animals of these high-output breeds are more prone to disease and productivity rapidly declines. Already, some traditional breeds are required for less intensive systems, and they often have inherent disease resistance [55], which will become important in the future when antibiotics are not available. 

## 3. Options for Farmers to Address Animal Welfare Challenges

The following are not necessarily solutions to the problems raised above but they offer some options for farmers to maintain viable enterprises and to continue the essential work that they perform in feeding the world’s humans.

### 3.1. Improved Nutrition

Addressing the nutrition challenges presented above requires farmers to reduce the nutrient density of their livestock diets. This will return livestock to a more natural diet and reduce the risk of metabolic disease. For example, to address increasing temperatures and the increased risk of heat stress, transferring ruminant livestock to a high fibre diet will reduce feed intake and the heat production associated with high-energy diets [56]. They are then less likely to suffer from heat stress. Similarly, converting intensive poultry production from grain-based housed production to a free-range system has the potential to utilise local forage resources and improve welfare, not just in temperate regions where this transition is most common but also in the tropics [57]. Alternative crops can be considered which cope better with increased temperatures, drought, and extremes. One such crop is amaranth, a pseudocereal with good amino acid balance and respectable oil, mineral, and antioxidant contents, provided it is heat-treated to negate antinutritive compounds, including tannins, phytic acid, saponins and others, in the grain [58]. 

Replacing antibiotics in the diet of chickens with feed additives that stimulate the birds’ immune system is also possible. Flavonoids and other active ingredients that do this are plentiful in a range of herbs and spices [59]. Some of the most promising supplements are olive oil byproducts, olive leaves, and alfalfa, but the optimum doses for most of the supplements have not yet been reliably determined. Other nutritional strategies that have been tried with considerable success include pre-, pro-, and synbiotics and antimicrobials such as lactoferrin that work directly on the microbes [60]. There are other feed additives that can have beneficial effects in controlling disease, such as sumac berries and thyme, but they are not available in sufficiently large quantities and, to make a major contribution, many farmers would have to transition to growing them [61]. Green tea is potentially available in large quantities and has beneficial antibacterial properties to potentially control coccidiosis without using coccidiostat antibiotics [62]. 

### 3.2. Diversification

Farmers on fertile land with good climatic conditions have to be cautious about any long-term investment in high-welfare facilities, such as a free-range system for pigs or chickens, when the price of grain crops seems likely to escalate and may, in future, be more profitable than livestock farming. Similarly, livestock farmers with intensive production facilities should be mindful of escalating grain prices when considering their future [63]. How restrictions of nutrients affect the welfare of livestock that have been bred for high productivity is not clear. However, it is likely that the welfare consequences are serious, including the risk of subclinical metabolic diseases and a greater susceptibility to infectious diseases. Some welfare benefits are possible, avoidance of ruminal acidosis in cattle, for example, and reduction in heat stress.

Many farmers are considering diversifying away from livestock production because markets for artificial replacers for meat and milk are developing rapidly. Although currently operating as a niche market, the growth has been strong, especially for milk substitutes. Most people are willing to try these alternative protein sources [64], and people will consider replacing meat and or milk totally if suitable substitutes are available.

Government intervention to assist farmers to produce what is most beneficial for public health and longevity is desirable. This may take the form of instruction to limit meat consumption and/or purchase of agricultural products that are most beneficial for the environment. For example, in 2016, the Chinese government advised its citizens to reduce meat consumption by 50% [65].

### 3.3. Silvopastoral Systems

Combining trees and agricultural crops has many advantages [66] but they are not easily integrated into intensive livestock production systems. For the small producer of extensively kept livestock, especially cattle and sheep, this can lead to better water usage, improved livestock welfare, more fertile land for the trees, and increased disease resistance in the livestock due to a reduced stocking density. Replacing grass with legumes within the tree stand will potentially reduce or eliminate the need for nitrogen fertiliser [67]. In the EU, there is particularly strong support for such silvopastoral systems [68], which will improve animal welfare. Similarly, the support for soils, plants, rare livestock breeds, and ecosystems in the EU may improve animal welfare indirectly, for example, by increasing plant diversity and allowing grazing animals to self-medicate.

### 3.4. Reversing Genetic Gains

The emphasis for the last 50 years has been for farmers and their associated breeding companies to develop breeds of livestock that grow faster, have higher output, and get sick less often. Although reduced disease equates with better welfare, production-related diseases, such as lameness and mastitis in dairy cows, have not been able to be controlled because of increasing output. Some certification schemes now require farmers to use breeds of livestock and, in particular, chickens that are not fast-growing [69]. This will help to control production-related diseases. In the future, limited feed supplies for intensive livestock production may mean that such breeds are more suitable and, therefore, these breeds should be preserved before they become extinct. 

### 3.5. Precision Livestock Farming

The use of artificial intelligence and other technology in livestock production offers the potential to improve livestock welfare [70]. But it may also herald the day when few people are involved in day-to-day contact with the animals, and empathy cannot be assumed in modern technology. Improvements in welfare could result from the elimination of traditional barbed wire fences, which are also harmful to the welfare of wildlife, and replacement of electronic fences with E-collars. However, it must be remembered that not all livestock can adjust to control by electric shocks in just the same way that not all dairy cows adapt to robotic milking. Although the welfare of all animals is important, it is inevitable that there will be continued selection for livestock that are easily managed by machines. Drones have the potential to be used for monitoring grazing stock, water troughs, feed availability, and grazing patterns as well as their use in mustering in rangelands. Robotic milking is becoming more and more widespread and robotic feeding of penned calves is now also well established. Large intensive units with thousands of animals are most likely to benefit from automatic detection of animal welfare. Poultry, pig, and now dairy are increasingly adopting this method of production as worldwide demand for the products increases. The availability of new technology for collecting and processing information from cameras and microphones is likely to be useful for farmers to assess the welfare of their animals remotely. One of the most promising indicators of welfare is the animals’ vocalisations [71]. An individual animal’s welfare can be determined from the call characteristics, usually measured by the valency and intensity of the call. Pitch, volume intensity, harmonic-to-noise ratio, tonality, duration, and repetition frequency all vary with the different types of call that relate to the animal’s situation. However, although this can be determined for individual animals, research to extract the welfare-related characteristics from the combined calls of large groups of animals is only just starting. Deep-learning algorithms are being developed to extract the relevant call characteristics in the large groups of intensively kept domestic animals. Conceivably, the pitch (frequency) and volume will be the best indicators, being easier to discern from group calls than other characteristics. 

Analysis of bioacoustics signals on a farm could potentially alert farmers to the presence of an intruder, a delay in the provision of feed, social disruption in group members, the accumulation of noxious gases, and other hazards in intensive units. In dairy cow herds, cows in partition emitting vocalisations associated with pain could be detected. 

These are just a few examples of ways in which livestock welfare could be improved by using advanced technological methods on farms. There are many others, but the cost is often high and farmers’ trust in such technology is limited. In addition, farms are highly diverse and it will be difficult to produce management aids that suit all types of farms. The success of AI to improve livestock welfare will be dependent on the development cost, the profit from each unit, and the size of the market.

## 4. Conclusions

Livestock farmers currently face unprecedented challenges that will affect the welfare of their livestock, mostly not of their own making. This includes climate change, disease challenges, labour issues, changing demand, and rapidly rising and volatile prices of key resources that will make it hard to sustain a good return from their investment over many years. This is likely to discourage them from making major investments in livestock production, many of which would improve welfare, and some will turn to alternative enterprises. Some strategies are available to reduce risks to their enterprises but not all farmers will escape hardship, and this may impact on the welfare of their livestock.

## Data Availability

No new data were created or analyzed in this study. Data sharing is not applicable to this article.

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
