# Peer review of "Farm Animal Welfare—From the Farmers’ Perspective"

_animals, 2024, doi:10.3390/ani14050671_

Round 1

Reviewer 1 Report

Comments and Suggestions for Authors

This is a well-written, personal exploration of pressures on livestock farmers attempting to reconcile animal welfare concerns with financial constraints and problems arising from urbanisation and climate change. -as viewed from the southern hemisphere.

The arguments are sound so far as they go. They do not consider the duty of farmers to the land -the soil and the plant and animal ecosystem sustained by the soil.  This is a high priority concern in Europe. I recognise that this was not in the address which forms the basis for this paper but it may merit inclusion, together with the obligation for society/governments to provide farmers with public money for public goods.

Minor points

l88-89 the prediction equation for water intake includes a linear increase with increasing air temperature.. Surely this should only apply above the UCT 9ot thereabouts}

l117 substitute 'less concern' for 'diminished responsibility'

l259 please don't write decimated. This describes a 10% cull.

Author Response

This is a well-written, personal exploration of pressures on livestock farmers attempting to reconcile animal welfare concerns with financial constraints and problems arising from urbanisation and climate change. -as viewed from the southern hemisphere.

The arguments are sound so far as they go. They do not consider the duty of farmers to the land -the soil and the plant and animal ecosystem sustained by the soil.  This is a high priority concern in Europe. I recognise that this was not in the address which forms the basis for this paper but it may merit inclusion, together with the obligation for society/governments to provide farmers with public money for public goods.

Author’s response: Thank you for your comments. I have deliberately emphasized that economics are important to farmers in their attempts to improve welfare, which I believe is often not understood. Regarding the soil/plant/ecosystem nexus, I have included a short comment on this in the section on silvopastoralism, because preserving a healthy soil/plant/ecosystem, as recognised in Europe, may improve farm animal welfare, for example by providing a diversity of plants in the diet of grazing livestock.

“In the EU there is particularly strong support for such silvopastoral systems [68], which will improve animal welfare. Similarly, the support for soils, plants, rare livestock breeds and ecosystems in the EU may improve animal welfare indirectly, for example, by increasing plant diversity and allowing grazing animals to self-medicate.”

Marie-Laure Augère-Granier 2020. Agroforestry in the European Union. European Parliamentary Research Service, PE 651.982.

www.europarl.europa.eu/RegData/etudes/BRIE/2020/651982/EPRS_BRI(2020)651982_EN.pdf. Accessed 11 February, 2024.

Minor points

l88-89 the prediction equation for water intake includes a linear increase with increasing air temperature.. Surely this should only apply above the UCT 9ot thereabouts}

Author’s response: This has been removed at the request of another reviewer. As a point of interest, to the best of my knowledge, this degree of precision is not yet attainable. What you say is logical, but so far the equations are not yet able to demonstrate a UCT statistically. Of course, there are many factors affecting UCT, so no temperature is applicable to all animals and I give a range of 25-35oC.

l117 substitute 'less concern' for 'diminished responsibility'

Author’s response: changed

l259 please don't write decimated. This describes a 10% cull.

Author’s response: changed to ‘seriously affected’. 

Reviewer 2 Report

Comments and Suggestions for Authors

GENERAL COMMENTS: the author has presented a critical text about the welfare of farm animals from the farmers' point of view. The author's proposal is interesting because it is necessary to bring together the perspectives of the various stakeholders in the production chain to make the solutions more realistic. However, even knowing this is the Animals' Comment section, I expected to see arguments with a solid scientific basis centred on farm animals. I found gaps in this context and suggested a substantial improvement to make the text suitable for publication. Below are my comments.

- I noticed that the author gets lost in the discourse at several points, blurring the focus on the animal. According to Broom (1991), welfare is inherent to the individual. Therefore, a good approach is to establish the animal as the centre of discussions and the production chain around it, not the other way around.

- An example of this is in the Abstract. The issue of heat stress (L. 26-28) here is focussed on other points, such as forage and the concern for feed efficiency. There's no way to assess welfare from a utilitarian viewpoint alone. Heat stress causes physiological disorders, suffering and distress for the animal. In this way, I see how to discuss welfare with an eye to the animal's quality of life (a life worth living). A utilitarian view only aims to treat the animal well so that it offers what I want (or what capital needs to circulate as a product).

- The issue of skilled labour (L. 31) is directed at the worker, the concern about the lack of family succession. However, qualification aims to improve human-animal interaction, which is problematic today when we bring our assessment to animal production.

- L. 84-88: unnecessary sentence whose equation is not directly linked to the topic.

- L. 94 - About summer: we know today that the concern is about heat waves. Heat stress is not only contained in summer. In regions with tropical and subtropical climates, heat comes in the spring.

- L. 116-117 - "in developing countries because of these countries' need for foreign currency and diminished responsibility for animal welfare and environmental pollution"- I cannot agree with the sentence quoted by the author. This is untrue, as the organisations that import these products audited farms and livestock industries. Therefore, there is a lot of concern today about pollution and animal welfare in these countries.

- L. 128-129 - remove this part, as it is out of context with the manuscript.

- L. 135 - it is not animal welfare that suffers, but the animal, resulting in poor welfare.

- Item 2.4 - remove this item, as much of the text does not correspond with the aim of the manuscript. Only the part about the five domains could be kept but moved to the beginning of the text.

- Items 2.5, 2.6, 3.2 - again, texts that move away from the animal as the centre of discussions on animal welfare.

- Item 3.3 - I was unsure whether the author's intention was to highlight strengths or weaknesses. In any case, it needs more grounding so that it doesn't just remain the author's opinion.

- Item 3.4 - Rethink this topic. I don't think reversing animal genetic breeding is the way forward (not least because much of the progress in productivity is due to animal breeding and genetics), viewing production systems and seeing the animal as a sentient being (this is widely described in the literature).

- Item 3.5 - Rethink this topic. It may even be helpful for smaller herds, up to 1,000 - 5,000 heads. But when we bring the discussion to 30,000 animals, putting up a virtual fence becomes a considerable challenge. These are steps to be taken, but they shouldn't be the only ones.

Author Response

GENERAL COMMENTS: the author has presented a critical text about the welfare of farm animals from the farmers' point of view. The author's proposal is interesting because it is necessary to bring together the perspectives of the various stakeholders in the production chain to make the solutions more realistic. However, even knowing this is the Animals' Comment section, I expected to see arguments with a solid scientific basis centred on farm animals. I found gaps in this context and suggested a substantial improvement to make the text suitable for publication. Below are my comments.

I noticed that the author gets lost in the discourse at several points, blurring the focus on the animal. According to Broom (1991), welfare is inherent to the individual. Therefore, a good approach is to establish the animal as the centre of discussions and the production chain around it, not the other way around.

Author’s response: thank you for this perspective. In this Commentary, I have tried put myself in the position of the farmer, to think as they do, after five decades of working closely with them. Thus, I believe the approach is one that they would adopt, not necessarily always centred on individual animals, because sometimes, such as with chickens, their approach to welfare is more centred on the population than the individual animal. As you say, this is important because the farmers are key stakeholders in animal production; I have emphasized this at several points in the manuscript.

- An example of this is in the Abstract. The issue of heat stress (L. 26-28) here is focussed on other points, such as forage and the concern for feed efficiency. There's no way to assess welfare from a utilitarian viewpoint alone. Heat stress causes physiological disorders, suffering and distress for the animal. In this way, I see how to discuss welfare with an eye to the animal's quality of life (a life worth living). A utilitarian view only aims to treat the animal well so that it offers what I want (or what capital needs to circulate as a product).

Author’s response: This is a good example, thank you. In the abstract, I did point out that ‘climate change….exposes animals to heat stress’ I have now extended this to: ‘Heat-stressed animals have reduced welfare, and it is important to farmers that they convert feed to products for human consumption less efficiently, their immune system is compromised, and both the quality of the products and the animals’ reproduction are adversely affected.’ However, there is little or no information on the extent to which farmers can detect heat stress, and whether they would do anything if they can. I have added the following about farmer recognition and biological functioning: ‘Livestock experience heat stress, but unlike some other disorders, e.g. lameness (whay et al 2003), there is little or no information on the ability of farmers to recognise heat stress in their livestock, or whether they will act if they do detect it. Heat stress adversely affects animals’ biological functioning, but the extent of negative affect is unclear (Poisky and von Keyserlingk, 2017; Idris et al., 2024). High temperatures and droughts induced by anthropogenic climate change…’

- The issue of skilled labour (L. 31) is directed at the worker, the concern about the lack of family succession. However, qualification aims to improve human-animal interaction, which is problematic today when we bring our assessment to animal production.

Author’s response: I am sorry, I do not understand the point the reviewer is trying to make in the second sentence.

- L. 84-88: unnecessary sentence whose equation is not directly linked to the topic.

Author’s response: the equation has been removed.

- L. 94 - About summer: we know today that the concern is about heat waves. Heat stress is not only contained in summer. In regions with tropical and subtropical climates, heat comes in the spring.

Author’s response: summer changed to ‘hot periods of the year’.

- L. 116-117 - "in developing countries because of these countries' need for foreign currency and diminished responsibility for animal welfare and environmental pollution"- I cannot agree with the sentence quoted by the author. This is untrue, as the organisations that import these products audited farms and livestock industries. Therefore, there is a lot of concern today about pollution and animal welfare in these countries.

Author’s response: ‘diminished responsibility’ is changed to ‘less concern’. Another reference has been added, which compares concern for animal welfare in a variety of Eurasian countries. This demonstrates much less concern in some Asian developing countries. There are also many studies that demonstrate that the concept of animal welfare is generally not even understood in China, which can be added if necessary. ‘Environmental pollution’ is removed as it is not relevant.

- L. 128-129 - remove this part, as it is out of context with the manuscript.

Author’s response: In attempting to understand the farmers’ perception of animal welfare, it is important to consider who are most vulnerable. Although I was unable to find much evidence, this is important information about the exposure of smallholders that I would not want to ignore. I agree that the relevance to animal welfare was not clear, so I have extended this sentence so that it is evident: ‘and often have few spare resources, with the result that their feed stocks and animal welfare are likely to be most at risk.’.

- L. 135 - it is not animal welfare that suffers, but the animal, resulting in poor welfare.

Author’s response: Sorry, ‘suffers’ was not meant in the literal sense, but I see the potential for confusion. It and has been replaced by ‘deteriorates’.

- Item 2.4 - remove this item, as much of the text does not correspond with the aim of the manuscript. Only the part about the five domains could be kept but moved to the beginning of the text.

Author’s response: this section attempted to tackle the difficult area of the farmer/consumer interface. However, there was almost no information available, so I have removed this section.

- Items 2.5, 2.6, 3.2 - again, texts that move away from the animal as the centre of discussions on animal welfare.

Author’s response:

2.5 (now 2.4) I agree that the relevance of volatile fertilizer prices was not immediately obvious, so I have added two sentences at the start of this paragraph: ‘Fertilisers have been applied to animal feed crops at high rates for several decades in an attempt to maximise the carrying capacity of farmers’ land. Any threat to this high output of feed crops risks undernutrition of the farmers’ livestock, and consequently poor welfare.’

2.6 (now 2.5) I agree the reason for discussing antibiotics was not evident until the second half of this section, so I have reversed the order, putting the second half first.

3.2 (now 3.1) I have added sentences to maintain the focus on animal welfare.

- Item 3.3 - I was unsure whether the author's intention was to highlight strengths or weaknesses. In any case, it needs more grounding so that it doesn't just remain the author's opinion.

Author’s response: (now section 3.2) I have revised this section to make the implications for animal welfare abundantly clear.

- Item 3.4 - Rethink this topic. I don't think reversing animal genetic breeding is the way forward (not least because much of the progress in productivity is due to animal breeding and genetics), viewing production systems and seeing the animal as a sentient being (this is widely described in the literature).

Author’s response: The first sentence is a statement of fact and hardly needs any citation. Similarly, the second is also factual, supported by a reference. I have revised the remaining sentences to more accurately reflect the commentary intended.

- Item 3.5 - Rethink this topic. It may even be helpful for smaller herds, up to 1,000 - 5,000 heads. But when we bring the discussion to 30,000 animals, putting up a virtual fence becomes a considerable challenge. These are steps to be taken, but they shouldn't be the only ones.

Author’s response: (now 3.4) I have added a caveat at the end and somewhat modified the remaining content: ‘In addition, farms are highly diverse, and it will be difficult to produce management aids that suit all types of farms. The success of AI to improve livestock welfare will be dependent on the development cost, the profit from each unit and the size of the market.’

Reviewer 3 Report

Comments and Suggestions for Authors

The title, objectives and conclusions are not in agreement. They need improvement (e.g. Conclusions are based on just economics). 

2.7 Transport: It is not only the long transport that induce welfare problems, but also short transports, markets, control (assembly) posts and lairage including ruminants, pigs (EFSA, 2023), poultry and sick animals. 

Farm to farm transport induce diseases (epidemic?). More and more farms are specialized: breeding, fattening, dairy, specie (sheep, salmon, beef, porc, broiler etc.) 

Author Response

The title, objectives and conclusions are not in agreement. They need improvement (e.g. Conclusions are based on just economics). 

Author’s response: In this Commentary, I have tried put myself in the position of the farmer, to think as they do, after five decades of working closely with them. Farmers run a business, which determines that economics are at the heart of their decision making. This is not often understood by animal welfare scientists and advocates. I have modified the Conclusions to broaden the introductory sentences so that they do not simply reflect financial aspects.

2.7 Transport: It is not only the long transport that induce welfare problems, but also short transports, markets, control (assembly) posts and lairage including ruminants, pigs (EFSA, 2023), poultry and sick animals. 

Author’s response: I have changed the heading and text to make it clear that it is not only long distance transport that threatens animals’ welfare.

Farm to farm transport induce diseases (epidemic?). More and more farms are specialized: breeding, fattening, dairy, specie (sheep, salmon, beef, porc, broiler etc.) 

Author’s response: This is a good point and I’ve added comment at the end of this section as follows: ‘As well as transportation to slaughter, farmers are increasingly aware that transport between farms can transfer diseases. This is increasingly common with the specialization of livestock farms for single purposes, e.g. producing youngstock.’

Reviewer 4 Report

Comments and Suggestions for Authors

In this article, Prof. Clive Philips predicted a series of challenges that the livestock industry is likely to encounter in the future, which could affect animal welfare. After that, in order to address these challenges, several countermeasures and recommendations are also presented. The perspectives and viewpoints in the article are forward-looking, which makes it suitable for livestock-related practitioners to understand and refer to in-depth. However, before the article is published, there are a few minor issues that I would like the author to consider whether or not to start a discussion.

1 Population growth issues

The increasing human population has been referred to in the manuscript about 5 times in all. Food consumption must match the population growth rate and include both plant and animal foods. and the ratio of these two types of food is also dynamic and influenced by some factors. With the balance of the economy and the improvement of production technology, if the total increment of food decreases, it will definitely slow down the rate of population growth; at the same time, changes in the structure of food will also affect the rate of population growth. This is because the market makes direct economic demands on the farmer, and the farmer is motivated to make a range of choices thereby affecting animal welfare. This also caters to your suggestion to reduce the proportion of grain feeds. But in line 107, “As the world's human population grows up until the end of this century, less cereal grain will be available for feeding to livestock, which is currently about 36% of total grain production [20].” It predicts a challenge of the year 2100, nearly 80 years later. This example, regardless of what model it is based on, actually reinforces the reader's concern about the future source of feed ingredients, which is not necessarily the case. It is therefore recommended that the underlying theory be used to guide the reader, rather than supported by nebulous predictions.

2 The effect of the cultures and civilizations

The social license part refers to differences in the assessment tools and rules of animal welfare based on geographical areas or countries. However, the fact is that both the proportion of animal products consumed and the specific criteria for animal welfare are related to the different attitudes of people towards animals, and in particular the religious and civilizational backgrounds may have a direct impact on the future regional direction of animal production and animal welfare, including the consumption of animal foodstuffs. At the same time, the level of human civilization will have a more profound impact on animal welfare than the development of laws and standards that you mention, because laws and standards can only regulate the lowest moral standards of human beings and reduce or eliminate the incidence of animal cruelty, whereas culture and civilization are the greatest driving force in optimizing animal welfare. So, are you considering complementing the benign effects of cultural and civilizational advances on animal welfare in animal husbandry in the future? This impact will undoubtedly diminish many of the original concerns.

3 Declining human capital

In lines 131-135, migration had been mentioned to decrease animal welfare for the problem of lacking skilled labor. It seems that it is a traditional point of view of the industrializing of farm animal production. Industrialization does not only mean a reduction in labor inputs, but in today's livestock production, it also means an increase in precision animal husbandry and intelligence, which makes it possible to produce livestock that takes into account the welfare of the animals and meets their needs. While the future of farming will still require farmers who understand the needs of their animals, the future of livestock production is likely to be characterized by the need for more skilled workers. So, the number of traditionally skilled farmers may decrease, but may not necessarily affect animal welfare.

Author Response

In this article, Prof. Clive Philips predicted a series of challenges that the livestock industry is likely to encounter in the future, which could affect animal welfare. After that, in order to address these challenges, several countermeasures and recommendations are also presented. The perspectives and viewpoints in the article are forward-looking, which makes it suitable for livestock-related practitioners to understand and refer to in-depth. However, before the article is published, there are a few minor issues that I would like the author to consider whether or not to start a discussion.

Author’s response:

1 Population growth issues

The increasing human population has been referred to in the manuscript about 5 times in all. Food consumption must match the population growth rate and include both plant and animal foods. and the ratio of these two types of food is also dynamic and influenced by some factors. With the balance of the economy and the improvement of production technology, if the total increment of food decreases, it will definitely slow down the rate of population growth; at the same time, changes in the structure of food will also affect the rate of population growth. This is because the market makes direct economic demands on the farmer, and the farmer is motivated to make a range of choices thereby affecting animal welfare. This also caters to your suggestion to reduce the proportion of grain feeds. But in line 107, “As the world's human population grows up until the end of this century, less cereal grain will be available for feeding to livestock, which is currently about 36% of total grain production [20].” It predicts a challenge of the year 2100, nearly 80 years later. This example, regardless of what model it is based on, actually reinforces the reader's concern about the future source of feed ingredients, which is not necessarily the case. It is therefore recommended that the underlying theory be used to guide the reader, rather than supported by nebulous predictions.

Author’s response: it is very hard to see how feed production can be maintained at the current rate with the growing human population and growing demand for livestock products. I did not think it was relevant to go into detail here about the improved efficiency of producing human food directly from plant material compared to livestock, but have done so in several other publications. Also, this did not seem to be the place for a discussion about cell grown meat. As this is a Commentary based on existing evidence, the reasonable assumption that livestock will increasingly be kept on marginal land seems well founded.

2 The effect of the cultures and civilizations

The social license part refers to differences in the assessment tools and rules of animal welfare based on geographical areas or countries. However, the fact is that both the proportion of animal products consumed and the specific criteria for animal welfare are related to the different attitudes of people towards animals, and in particular the religious and civilizational backgrounds may have a direct impact on the future regional direction of animal production and animal welfare, including the consumption of animal foodstuffs. At the same time, the level of human civilization will have a more profound impact on animal welfare than the development of laws and standards that you mention, because laws and standards can only regulate the lowest moral standards of human beings and reduce or eliminate the incidence of animal cruelty, whereas culture and civilization are the greatest driving force in optimizing animal welfare. So, are you considering complementing the benign effects of cultural and civilizational advances on animal welfare in animal husbandry in the future? This impact will undoubtedly diminish many of the original concerns.

Author’s response: this section has been removed.

3 Declining human capital

In lines 131-135, migration had been mentioned to decrease animal welfare for the problem of lacking skilled labor. It seems that it is a traditional point of view of the industrializing of farm animal production. Industrialization does not only mean a reduction in labor inputs, but in today's livestock production, it also means an increase in precision animal husbandry and intelligence, which makes it possible to produce livestock that takes into account the welfare of the animals and meets their needs. While the future of farming will still require farmers who understand the needs of their animals, the future of livestock production is likely to be characterized by the need for more skilled workers. So, the number of traditionally skilled farmers may decrease, but may not necessarily affect animal welfare.

Author’s response: thank you for this important comment, which I completely agree with. I have added a paragraph at the end:

‘With the industrialization of livestock farming, there is a need for the labour force to be increasingly skilled. An important asset of skilled labour is the ability to detect behavioural abnormalities in the livestock they care for. For example, dairy farmers may have difficulty detecting cows that are lame within their herd, which is usually by observation by farmers and their staff [13, 34]. The detection of lame cows is increasingly difficult as herd sizes increase, and staff have less time to detect abnormalities. Automated detection methods are being developed but their cost, effectiveness and a negative farmer perception has led to limited uptake [34]. Availability of veterinary attention also cannot be guaranteed as the costs of treating individual animals may exceed their value [35].’

Round 2

Reviewer 2 Report

Comments and Suggestions for Authors

An excellent scientific debate with the author. I would go further on some points, but to avoid making the review process tedious, I am satisfied with the state of the art. I suggest approving the manuscript.

Author Response

Thank you for the debate!

Reviewer 3 Report

Comments and Suggestions for Authors

Author Response

Thank you for your approval of the revision.